# A Comparative Study of Three Approaches to Fibre’s Surface Functionalization

**DOI:** 10.3390/jfb13040272

**Published:** 2022-12-02

**Authors:** Judyta Dulnik, Oliwia Jeznach, Paweł Sajkiewicz

**Affiliations:** Laboratory of Polymers and Biomaterials, Institute of Fundamental Technological Research Polish Academy of Sciences, Pawińskiego 5b, 02-106 Warsaw, Poland

**Keywords:** surface activation, functionalization, electrospun fibres, hydrolysis, plasma, aminolysis

## Abstract

Polyester-based scaffolds are of research interest for the regeneration of a wide spectrum of tissues. However, there is a need to improve scaffold wettability and introduce bioactivity. Surface modification is a widely studied approach for improving scaffold performance and maintaining appropriate bulk properties. In this study, three methods to functionalize the surface of the poly(lactide-co-ε-caprolactone) PLCL fibres using gelatin immobilisation were compared. Hydrolysis, oxygen plasma treatment, and aminolysis were chosen as activation methods to introduce carboxyl (-COOH) and amino (-NH_2_) functional groups on the surface before gelatin immobilisation. To covalently attach the gelatin, carbodiimide coupling was chosen for hydrolysed and plasma-treated materials, and glutaraldehyde crosslinking was used in the case of the aminolysed samples. Materials after physical entrapment of gelatin and immobilisation using carbodiimide coupling without previous activation were prepared as controls. The difference in gelatin amount on the surface, impact on the fibres morphology, molecular weight, and mechanical properties were observed depending on the type of modification and applied parameters of activation. It was shown that hydrolysis influences the surface of the material the most, whereas plasma treatment and aminolysis have an effect on the whole volume of the material. Despite this difference, bulk mechanical properties were affected for all the approaches. All materials were completely hydrophilic after functionalization. Cytotoxicity was not recognized for any of the samples. Gelatin immobilisation resulted in improved L929 cell morphology with the best effect for samples activated with hydrolysis and plasma treatment. Our study indicates that the use of any surface activation method should be limited to the lowest concentration/reaction time that enables subsequent satisfactory functionalization and the decision should be based on a specific function that the final scaffold material has to perform.

## 1. Introduction

Polymers are frequently used in tissue engineering applications due to the ease of adjusting their mechanical properties, fabrication to the desired shape, and the possibility of chemical modification [1]. Among them, aliphatic polyesters play an important role as they degrade chemically by hydrolysis, or less often, enzymatic processes without toxic products [2,3]. Lactide, caprolactone, and glycolide are the commonly used monomers to synthesise aliphatic polyesters and copolymers that are applied in tissue engineering [4]. Polymers can be manufactured using various techniques, e.g., electrospinning [5], 3D printing [6], and freeze-extraction [7]. Their bulk properties can be tailored via various methods, e.g., copolymerization [8], controlled crystallization [9], or the use of additives [10]. Thanks to the possibility of properties modification, polyester-based scaffolds are of research interest in the regeneration of a wide spectrum of tissues including skin [11], blood vessels [12], neural tissue [13], bladder tissue [14], bone [15], tendon [16], etc. 

However, the functionality of polyester scaffolds in tissue engineering applications is limited by their surface properties—hydrophobicity and lack of bioactivity [3].

A few approaches are implemented to introduce missing bioactivity: using a decellularized natural extracellular matrix (ECM) as a scaffold [17,18,19], fabrication of scaffold from natural polymers containing adequate peptide sequences [20,21,22], or manufacturing synthetic-natural polymer composites [23,24,25]. Nowadays, a lot of attention is devoted in particular to the studies on biomolecule immobilisation on the surface of synthetic polymers to impart bioactivity and take advantage of their suitable mechanical properties and manufacturability [3].

It is well-known that cells are connected with the ECM via their receptors—integrins that recognize special peptide sequences. Up to now, a lot of sequences were found in the so-called cell adhesion molecules [26]. The accessibility of peptide sequence for given cell receptors is determined by the conformation of individual proteins and modulated by immediately adjacent amino acids [27]. One of the most examined is the Arginyl-glycyl-aspartic acid (RGD) motif, which is present mainly in fibronectin and the denatured form of collagen. RGD sequence present in gelatin is recognized by various types of cells, and in the case of L929 fibroblast cells via α5β1 and αvβ3 integrins [28]. Gelatin attachment to fibrous materials made of aliphatic polyesters has been shown to improve cell attachment and spreading [29].

The simplest method of surface modification by attaching a bioactive molecule like gelatin is physisorption, which is conducted by immersing the material in a biomolecule solution. In this case, electrostatic interactions, hydrogen bonds, van der Waals forces, and hydrophobic interactions are responsible for binding biomolecules with the surface [30]. Another physical approach is a layer-by-layer technique, which consists of depositing alternating layers of oppositely charged materials [31]. Mild conditions, simplicity, and applicability to a lot of materials are advantages of physical methods [23]. However, weak interactions could result in the instability of the coating. A second concern is the loss of functionality due to the conformational changes of a biomolecule, especially caused by hydrophobic interactions [32]. Chemical immobilisation is an alternative way to introduce bioactive molecules on the scaffold surface. The first step is the addition of functional groups on the surface to provide the binding site with a biomolecule. It is achieved by various methods, e.g., wet chemical methods, such as hydrolysis and aminolysis [33], photographing of poly(acrylic acid) or poly(methacrylic acid) [34], thiol bonding [35], and azide-alkyne click chemistry [36], etc. The main benefit of this kind of functionalization is a stable coating provided by covalent bonding between polymer and biomolecule. However, it should be taken into consideration that some of these methods could be destructive to materials [37,38] and by-products could exhibit cytotoxicity [39], so the method optimization toward specific materials is required.

Among the variety of biodegradable polymers commonly used in tissue engineering, poly(lactide-co-ε-caprolactone) PLCL has been reported to be the most promising candidate for tissue engineering. PLCL surpasses aliphatic polyester-like poly(L-lactic acid) (PLLA) and poly(ε-caprolactone) (PCL) in some of their properties. The most important disadvantages of PLLA and PCL are related to mechanical properties—PLLA and PLGA exhibit high strength and poor toughness, whereas PCL has good toughness but low strength [40,41,42].

By the association of PLLA with PCL into PLCL copolymer the advantages of both PLLA and PCL are obtained together with compensation; both the brittle behaviour of PLLA and the low stiffness of PCL [43,44] as well as the local acidification reduction due to the degradation of PLLA [45]. PLCL also offers an adjustable degradation rate adapted to its use in tissue engineering, tailorable by changing the PCL/PLLA ratio [46]. A strong shape–memory effect has also been observed for PLCL, depending on PCL/PLLA ratio [47].

In general, it may be concluded that PLCL is the most suitable polymer for various tissue engineering applications, particularly when adjustable elasticity and degradability are required. For instance, a braided PLCL scaffold (PLLA/PCL ratio  =  85/15) has been recently designed for ligament tissue engineering [48].

Therefore, the main objectives of this work were to examine surface activation methods such as hydrolysis, oxygen plasma, and aminolysis acting as a first step in functionalising a PLCL nonwoven material with gelatin and to compare the resulting material’s properties. Our goal was to provide comprehensive data that could be useful to many people while deciding what method to use for their specific needs. Since gelatin has both -NH_2_ and -COOH groups in abundance [49], in order to form a covalent bond between gelatin and fibres’ surface, grafting methods such as 1-ethyl-3-(3-dimethylaminopropyl)carbodiimide hydrochloride/N-hydroxysuccinimide (EDC/NHS) coupling and glutaraldehyde crosslinking were used. Additionally, materials after physical entrapment of gelatin and immobilisation using EDC/NHS coupling without previous activation were prepared as controls. All the approaches were compared for their immobilisation effectiveness and stability, impact on fibres morphology, wettability, chemical structure, molecular weight, functionalization stability as well as mechanical properties and L929 cells response.

## 2. Materials and Methods

### 2.1. Materials

PLCL 70:30 (Resomer^®^ LC703S, inherent viscosity = 1.3–1.8 dL/g, T_g_ = 32–42 °C) was purchased from Evonik (Essen, Germany). Hexafluoroisopropanol (HFIP) (purity degree 98.5%) was procured from Iris Biotech GmbH (Marktredwitz, Germany). Ethylenediamine (EDA) (purity degree 99.5%), sodium hydroxide (NaOH), isopropanol (purity degree 99.8%), glutaraldehyde (GTA, conc. 25%), and tetrahydrofuran (THF) were purchased from Chempur (Piekary Śląskie, Poland). Gelatin (G1890, type A, gel strength ~300 g Bloom) and bicinchoninic acid (BCA) assay kit (QuantiPro™, for 0.5–30 µg/mL protein conc.) was supplied by Sigma-Aldrich (St Louis, MO, USA). N-hydroxysuccinimide (NHS), 1-ethyl-3-(3-dimethylaminopropyl)carbodiimide hydrochloride (EDC), Dulbecco’s Modified Eagle Medium (DMEM), Phosphate-Buffered Saline (PBS) pH 7.4, Foetal Bovine Serum (FBS), Trypsin EDTA, antibiotic (Penicillin-Streptomycin 10,000 U/mL), Presto Blue, NucBlue, and ActinGreen were purchased from Thermo Fisher Scientific (Waltham, MA, USA). Mouse fibroblast L929 were purchased from Sigma-Aldrich (ATTC).

### 2.2. Electrospinning

All materials were electrospun using a Fluidnatek LE-50 electrospinning machine by Bioinicia (Valencia, Spain). The polymer solution was prepared by dissolving PLCL in HFIP with 7% *w*/*w* polymer concentration and stirred for 24 h at room temperature. Two needles were fed with polymer solution at a rate of 3 mL/h each and the voltage was kept in a 11–13 kV range. A drum collector (ø 5 cm) was rotating at 300 RPM speed and the distance from a needle tip to a collector’s surface was set to 15 cm. Electrospinning took place in monitored air conditions set for 38 °C temperature and 40% humidity. Temperature elevated to the polymer’s glass transition temperature (as provided by the manufacturer T_gPLCL_ 32–42 °C) was used in order to avoid shrinking that happens after electrospinning of PLCL nonwoven at room temperature (20–23 °C).

After electrospinning was completed, to make sure no residual solvent was present in the fibres, nonwoven mats were kept under a fume hood overnight and then transferred to a vacuum oven set to 50 mBar for 24 h at room temperature. All surface activation/crosslinking/functionalization steps were carried out on 4.5 cm × 4.5 cm squares that electrospun materials were cut into.

### 2.3. Optimization of Surface Activation Methods

In order to determine the optimal parameters for the chosen surface activation methods, each of them was performed with a set of varying conditions (Table 1). All samples were then subjected to a series of tests to assess their morphology, physicochemical properties, and activation efficiency. This approach enabled the selection of a final set of samples with an already activated surface to undergo the next steps of functionalization.

#### 2.3.1. Hydrolysis

Alkaline hydrolysis experiments were performed at room temperature (25 °C). Samples were submerged in NaOH solutions with concentrations 0.05 M, 0.1 M, 0.25 M, 0.5 M, and 1 M for 10 min, 30 min, 60 min, and 180 min as mentioned in Table 1. An additional sample underwent a reaction in 1 M NaOH solution for 360 min to determine changes that occur in material subjected to extreme conditions. After appointed reaction times, samples were transferred to 0.01 M HCl solution for protonation of newly created COO- groups and subsequently to demineralised water for rinsing. Thus prepared samples were dried in a vacuum oven at room temperature. Because of the observed decrease in thickness of samples treated with higher NaOH concentrations, the samples that were chosen for functionalization were also subjected to weight and thickness measurements before and after the surface activation step.

#### 2.3.2. Plasma

Cold oxygen plasma surface activation process was performed using a Plasma Cleaner (Diener, ZEPTO PCCE, Ebhausen, Germany) machine. All samples underwent plasma treatment with the same, low—1%, of instrument’s power, with 0.1 mBar oxygen pressure, for a broad range of times: 5 s, 10 s, 15 s, 25 s, 40 s, and 60 s as mentioned in Table 1.

#### 2.3.3. Aminolysis

This part of the experiments was based on the data collected in our previous study on aminolysis, where an increase in the amount of NH_2_ groups with the diamine concentration and reaction time was observed and described [50]. Three concentrations—2%, 6%, and 10% *w*/*v*, and four different times—5 min, 10 min, 15 min, and 30 min were examined, as mentioned in Table 1. For this work, to provide various amounts of NH_2_ groups on the materials’ surface for the functionalization study, three different reaction conditions were chosen that result in an amount of free NH_2_ groups on the surface equal to 1.42 ± 0.08 × 10^−8^ mol/mg, 5.43 ± 0.90 × 10^−8^ mol/mg, and 8.63 ± 1.16 × 10^−8^ mol/mg for A1, A2, and A3 samples, respectively. Aminolysis was conducted by immersing samples in ethylenediamine/isopropanol solution, the temperature of the reaction was set at 30 °C. The process was conducted in an orbital shaker incubator. After aminolysis samples were washed three times in deionized water and dried under a vacuum overnight.

### 2.4. Functionalization

Gelatin immobilisation was executed via a three-step approach. First, PLCL nonwoven materials cut into 4.5 cm × 4.5 cm squares were subjected to one of the three surface activation processes: hydrolysis, cold oxygen plasma, or aminolysis. The final set of process parameters was chosen for each method based on the results of the optimization step: hydrolysis—0.1 M, 0.25 M, 0.5 M, and 1 M. for 3 h, for plasma—10 s and 40 s of treatment, for aminolysis—three sets of EDA concentration and reaction time: 2%—5 min, 2%—10 min, and 6%—5 min on the basis of the results from a previous study [50].

The next step consisted of gelatin crosslinking. Samples after surface activation with hydrolysis and plasma were placed for 1 h in *w*/*w* 0.23%/0.12% EDC/NHS solution in ethanol:water mixture with at 7:3 ratio. Simultaneously, a set of samples after aminolysis were placed in 1% *w*/*v* GTA in a water solution for 2.5 h at 24 °C to compare cross-linking methods. All of the samples were thoroughly rinsed in demineralized water and right afterwards submerged in 0.2% gelatin solution for 20 h at 37 °C. Gelatin immobilisation was completed to make sure no unattached gelatin was present within the material, and samples were once again placed in demineralized water and stirred, this time for 24 h at 37 °C. Two control samples of PLCL without surface activation were prepared. One placed in an EDC/NHS solution followed by a gelatin bath, the other one only going through the last step. All functionalized sample names are specified in Table 2.

### 2.5. Functionalization

#### 2.5.1. Water Contact Angle

Water contact angle was measured with a goniometer (Data Physics OCA 15EC, Filderstadt, Germany) using the sessile drop method. For each sample, the water contact angle was measured not less than 10 times. For each measurement, the angle was recorded at 0.5 s, 3 s, 10 s, and 30 s after placing the demineralized water drop on the surface of a sample.

#### 2.5.2. Nonwovens Morphology

Scanning electron microscopy images (JSM-6010PLUS/LV InTouchScope™, JEOL, Tokyo, Japan) were taken of each sample in order to assess changes in their morphology after the surface activation process and then again after the functionalization step. Samples were sputter-coated with approx. 10 nm of gold.

#### 2.5.3. Quantification of -COOH Groups

Both plasma and hydrolysis treatments were expected to increase the -COOH group density on the material’s surface. Toluidine blue-O test was performed to assess how effective those processes were in various conditions. This colourimetric method works by dye-staining deprotonated carboxyl groups on fibres’ surfaces through ionic interaction. Here, an adapted routine proposed by Gupta et al. [51] was used. Briefly, a 0.5 mM solution of toluidine dye with pH10 was prepared. Samples were soaked in it for 5 h at 37 °C. Afterwards, nonwovens were thoroughly rinsed with NaOH of pH 9 to remove any unattached dye. Next, the samples were transferred to acetic acid 50% *w*/*w* solution to detach the dye in order to measure its concentration with a UV-vis spectrophotometer at 630 nm (The Multiskan™ GO, Thermo Fisher Scientific, USA).

#### 2.5.4. Fourier Transform Infrared Spectroscopy (FTIR)

A series of infrared attenuated total reflectance (ATR) spectroscopy tests were performed for two main purposes. First, during the optimization part of the work, to determine the changes that occur in the materials subjected to hydrolysis and plasma treatment, and later to assess the efficiency of gelatin attachment to the materials functionalized after surface activation with all three types of treatment. The infrared ATR spectroscopy tests were conducted with a spectrometer Bruker Vertex 70 (Mannheim, Germany).

#### 2.5.5. Molecular Weight

Molecular weight measurements were performed using a Nexera (Shimadzu, Japan) device with THF as eluent. Gel permeation chromatography (GPC) equipment consisted of a pump, column (Phenogel^TM^ 5 μm 10^5^ Å, Phenomenex), refractive index detector (RID-20A, Shimadzu), and LabSolutions GPC software. All samples were dissolved in THF in 2 mg/mL concentration by vortex stirring for 24 h at room temperature and then filtered with hydrophobic polytetrafluoroethylene (PTFE) syringe filters with 0.22 µm pore size. THF was used as a mobile phase with a flow rate equal to 1 mL/min. The temperature was set at 40 °C. The molecular weight results were calculated using a polystyrene standard (number averaged molecular weight from 3470 Da to 2,520,000 Da) calibration curve coupled with the Mark–Houwink equation where α = 0.7, K = 14.1 × 10^−5^ dL/g for polystyrene standard polymer, and α is 0.68, K is 4.84 × 10^−4^ dL/g for PLCL according to Nuuttilla M. et al. [52] were used.

#### 2.5.6. Mechanical Tests

Mechanical tests were conducted using a Lloyd EZ-50 (United States of America) device. Uniaxial tensile testing was performed with a 5 mm/min extension speed. From each type of material three 50 mm × 5 mm rectangular samples were cut, their thickness was measured at the ends and the results were averaged. For testing a sample was placed in between two rubber-lined clamps, leaving the 20 mm × 5 mm part of the sample to be extended.

#### 2.5.7. Functionalization Stability

In order to determine the stability of gelatin coating on fibres’ surfaces, all the samples underwent testing in conditions mimicking the environment of a living organism. Each sample (4.5 cm × 4.5 cm) was immersed in 100 mL of PBS with a pH of 7.4 and placed in an orbital shaker at 37 °C, 100 RPM (revolutions per minute). To prevent bacterial or fungi growth, sodium azide in a concentration of 0.1% was added. After 24 h and 7 days, samples were taken out and washed with demineralized water and dried with a vacuum oven.

#### 2.5.8. Quantification of Gelatin Amount on Fibres’ Surfaces

The amount of protein on fibres’ surfaces was measured both right away after functionalization as well as after biodegradation using a BCA assay kit (QuantiPro™, for 0.5–30 µg/mL protein conc.). For each of the modified materials, three samples of weight 1 ± mg were investigated. Each sample was put into an Eppendorf vial and 0.5 mL of deionized water with 0.5 mL of BCA solution was added to the vial. To obtain a calibration curve, gelatin solutions of concentrations in the range of 5–30 µg/mL were prepared. The samples, as well as calibration solutions, were incubated for 2 h at 37 °C. Absorbance was measured with a UV-vis spectrophotometer at 560 nm. Results were shown as the amount of gelatin on the surface in µg per mg of fibrous sample.

### 2.6. Functionalization

The mouse fibroblast L929 cell line that was used to perform all cellular tests was purchased from Sigma-Aldrich (ATTC). The culture medium used consisted of 89% Dulbecco’s Modified Eagle’s Medium, 10% foetal bovine serum, and 1% antibiotic (penicillin-streptomycin). For cytotoxicity on extracts and direct contact tests, nonwovens were cut into 5 mm and 10 mm discs, respectively, sterilised with 80% ethanol and UV light for 30 min for each side.

#### 2.6.1. Quantitative Tests: Cytotoxicity on Extracts and Viability Test in Direct Contact

For the cytotoxicity test, material extracts were obtained by placing 5 samples (5 mm in diameter) of each nonwoven type in a 96-well plate, then 200 µL of culture medium was added to each well, and afterwards the plate was kept at 37 °C and gently stirred for 24 h. At the same time, another 96-well plate was seeded with 10^4^ cells/well and placed for 24 h in an incubator (37 °C, 5% CO_2_). After 24 h the culture medium in the cell-seeded wells was replaced with material extracts and put in the incubator for another 24 h.

To assess cell viability in direct contact with the materials, for each time point (1, 2, 3 days), 3 samples (10 mm in diameter) were placed in separate 48-well plates and seeded with a density of 10^4^ cells/well and the plates were then put into the incubator. For the 0-day control time point, the same number of cells was seeded into 3 empty wells and the viability test was performed after 5 h so that cells had time to attach to the surface of the well bottom.

For both cytotoxicity on extracts and 0–3-day tests in direct contact, the Presto Blue method was used to assess cell metabolic activity. First, the culture medium was removed, then each well was washed with PBS, Presto Blue reagent was added with PBS in the 1:9 ratio and next the plate was put in the incubator for 40 min. After this time, 100 µL was transferred from each well to a 96-well plate and the fluorescence was read with the excitation/emission 530/620 nm filters.

#### 2.6.2. Qualitative Morphology Tests

For the purpose of assessing the morphology of cells cultured in direct contact with functionalized nonwoven surfaces, two types of qualitative observation methods were employed. The first was scanning electron microscopy (SEM) imaging and the second was observing cells’ actin skeleton and nuclei with fluorescence microscopy. For these, a 5-day cell culture was performed where the materials were seeded with 2 × 10^3^ cells/well. Samples for SEM imaging were fixed by immersing 2.5% glutaraldehyde for 2 h and then dehydrated in a series of ethanol solutions (30–100%) followed by ethanol/hexamethyldisilazane (2:1, 1:2) mixtures. For fluorescence microscopy cells were fixed with 3% formaldehyde for 20 min, permeabilized with 0.01% triton × 100, and then stained with fluorescent dyes that attach to the cell actin skeleton (ActinGreen) and nucleus (NucBlue).

## 3. Results and Discussion

### 3.1. Optimisation of Surface Activation Methods

#### 3.1.1. Water Contact Angle

A nonwoven PLCL material, without any additional treatment, has a water contact angle of 131.43° ± 1.94°, which means it is hydrophobic which is typical for aliphatic polyesters [53]. The aminolysis reaction had no effect on this property as the samples A1, A2, and A3 had water contact angles equal to 131.31° ± 1.37, 130.74° ± 1.63°, and 131.28° ± 0.84°, respectively. Lack of wettability improvement or only a small change after aminolysis was reported earlier by others [54,55,56]. Both hydrolysis and plasma treatment lowered the water contact angle of PLCL samples.

Cold oxygen plasma proved to be an extremely time-efficient method of making PLCL fibres’ surface hydrophilic. All samples treated for at least 15 s had a 0° water contact angle only 10 s after the water drop was placed on its surface (Figure 1a). Because of the nature of fibrous materials, 0° means that the drop was effectively soaked into the material.

For hydrolysed PLCL samples, the decrease in water contact angle was dependent on both reaction time and NaOH concentration. Figure 1b,c shows a comparison of water contact angle measurement results for 180 min reaction time with varying NaOH concentrations (Figure 1b) and for 1 M NaOH solution used for reaction times from 10 up to 180 min (Figure 1c).

From the group of samples hydrolysed for 3 h, 0.25 M, or higher NaOH concentration was needed to make materials hydrophilic, with 0.5 M and 1 M NaOH enabling a water drop to be soaked into the material within only 3 s (Figure 1b). Looking from the other side, 1 M NaOH had to be used for at least 30 min for the water contact angle to drop below 100° (Figure 1c).

#### 3.1.2. Morphology

Fibres’ morphology of all samples after surface activation was compared to untreated PLCL (Figure 2a). All samples treated with oxygen plasma retained the material’s initial morphology whereas both hydrolysis and aminolysis did have an effect depending on activation conditions.

For hydrolysis, reaction times shorter than 3 h and concentrations lower than 0.5 M NaOH did not result in any fibres’ morphology alteration. A sample placed in 0.5 M NaOH for 3 h had its fibres’ surface slightly affected with small surface cracks (Figure 2b) and for the same time in 1 M concentration these changes were more pronounced as the top layer of the fibre surface seemed to be frayed, chipping off the fibres while the overall architecture is still intact (Figure 2c). The sample treated with 1 M NaOH for 6 h showed severe damage with both the surface of the fibres and the nonwoven structure being compromised (Figure 2d).

In addition, measurements of both weight and thickness before and after surface activation with NaOH (Figure 3) clearly show that the hydrolysis leads to surface degradation of the polymer severe enough that the thickness and weight of the nonwovens are significantly reduced. This reduction correlates linearly with NaOH solution concentration. However, for the low 0.1–0.25 M NaOH concentrations, the drop in both values did not reach 15%, and the samples that retained 72% and 46% of their initial weight, for 0.5 M and 1 M, respectively, were visibly thinner even for the naked eye.

A comparison of the fibre diameter distribution for the control PLCL sample and material treated with 1 M NaOH for 3 h (Figure 2a,c) shows how surface erosion caused by hydrolysis affected fibre thickness (Figure 4a,b). Whereas a bimodal character of the distribution for untreated PLCL is preserved in the hydrolysed sample, both of the local maxima present a lower value: 0.53 µm and 0.95 µm for control and 0.49 µm and 0.78 µm for NaOH treated material. A small percentage of fibres with a diameter below 0.3 µm are not present in the material after hydrolysis.

In the case of aminolysis surface activation, within the range of diamine concentration and reaction time that was chosen for this work based on our previous studies [50], no changes in fibres’ morphology were observed. It is valid even for treatment with 6% diamine solution for 5 min, which was the highest chosen reaction conditions (Figure 5a). Using higher diamine concentrations and longer reaction times led to fibres being prone to brittle cracking without changing fibre diameter (Figure 5b). This effect is known and utilized to fabricate fibrous fillers, e.g., for application in composite materials [57].

#### 3.1.3. The Efficacy of the -COOH Group Introduction to the Material’s Surface

Both alkaline hydrolysis and oxygen plasma treatment were meant to increase -COOH group content on the surface of PLCL material to enable functionalization with gelatin through crosslinking with EDC. Two tests were performed to assess this—Toluidine Blue O (TBO) staining and ATR-FTIR spectroscopy.

In the TBO experiment, a control untreated sample of PLCL nonwoven obtained an absorbance of 0.3 A.U. (absorbance unit). For the samples with surface activated with oxygen plasma (Figure 6a) this value increased with treatment time, reaching a plateau at 10 s time point with about 0.97 A.U. This result did not change in a significant way with longer exposure to plasma. With this method, the amount of -COOH groups on PLCL fibres’ surface can be, at maximum, increased to roughly 300% of its initial content.

The increase in -COOH groups detected on the surface of samples that underwent hydrolysis (Figure 6b) was, as predicted, relative to both NaOH concentration and reaction time. For 0.05 M and 0.1 M NaOH solutions the absorbance value increased to 0.4 A.U. after 10 min, it only reached about 0.6 A.U. even for the 3 h experiment. Much better results were achieved with solutions of higher NaOH concentrations for which the increase was more steady and steeper. For 0.5 M and 1 M NaOH solutions the absorbance measured were 7 and 10 fold, respectively, the untreated PLCL value.

For the same purpose of assessing -COOH group content on a material’s surface, FTIR spectroscopy analysis was focused on C=O bond stretching vibrations. A very apparent shift can be seen for the PLCL sample subjected to extreme (1 M NaOH, 6 h) hydrolysis conditions (Figure 7). Although such conditions were not included in the final scope of experiments planned for this work, they were useful in indicating a direction of changes that could be observed with FTIR spectroscopy. A peak at 1756 cm^−1^ that comes from the C=O bond that belongs to an ester group, typical for aliphatic polyesters such as PLCL, is replaced by a peak at 1721 cm^−1^ that comes from the C=O bond in the -COOH group [58]. The more hydrolysed the PLCL sample is, the clearer this shift. For a sample treated with 1 M NaOH for 1 h and 3 h, an elevation indicating a 1721 cm^−1^ peak is still apparent, whereas for a 0.5 M/3 h reaction the result is barely visible. Comparing these results to the TBO test it could be concluded that for a sample to show any shift in the FTIR spectrum it needs to reach at least 1.5 A.U in TBO staining. For the same reason, none of the FTIR spectra of the samples treated with oxygen plasma showed any differences in comparison to untreated PLCL material, which means this method is not as sensitive as the TBO test.

#### 3.1.4. Molecular Mass Change

Each of the three surface activation methods has a different effect on the polymer’s molecular weight as measured with gel permeation chromatography.

In the case of samples that underwent aminolysis and plasma treatment, a recognizable drop in molecular mass was observed. For the chosen set of reaction conditions, the plasma activation method proved to have a smaller impact on PLCL’s molecular weight than aminolysis. Oxygen plasma treatment also caused a more uniform decrease among both smaller and bigger polymer chains, whereas aminolysis induced more damage to molecules of smaller mass (Table 3, Figure 8a,b). A decrease in molecular weight after aminolysis and plasma treatment was observed by other authors [59,60].

In comparison to both of those methods, the surface activation with NaOH was much milder in its detrimental effect on the polymer’s molecular weight when looking at the main chromatography peak (Figure 9a). However, it is important to note that hydrolysed samples lost a fraction of their mass during the activation process (Figure 3) and thus the GPC result shows the molecular mass only of the sample remaining after the hydrolysis.

Although for the highest NaOH concentration of 1 M some shortening of longer polymer chains can be seen in Figure 9a, the overall changes for both M_w_ and M_n_ are minimal for the material fraction left after hydrolysis (Table 3). Interestingly, a new second peak for very small polymer molecules can be observed for samples treated with NaOH concentrations varying from 0.25 to 1 M (Figure 9b). The higher the NaOH concentration, the smaller the molecular weight of this new peak was. It is believed that this peak shows the residues of heavily severed polymer chains that constituted the outer layers of the fibres that were degraded during hydrolysis.

Combining SEM observation of rough and frayed fibres’ surface texture (Figure 2c), the mass and thickness loss of the samples (Figure 3) and a relatively small shift in molecular weight for the main chromatography peak, it was concluded that NaOH mainly affects the material’s surface, leaving the core part of the fibre unscathed. In comparison, both aminolysis and oxygen plasma inflict polymer chain severing in the whole mass of the sample. For a material with a substantial molecular mass reduction (M_w_ = 11.6 kDa, M_n_ = 6.5 kDa) [50], fibres are prone to break, rather than fray, as can be seen in the SEM image (Figure 5d) of a sample treated with a high diamine concentrated solution for a prolonged period of time.

### 3.2. Comparison of Functionalized Materials

#### 3.2.1. Gelatin Attachment and Stability

All samples after completed functionalization underwent FTIR spectroscopy as well as BCA staining to confirm a successful attachment and measure the amount of gelatin of their surface. BCA was used then again after 1 and 7 days of stability test.

A comparison of FTIR spectra in the 1500–1700 cm^−1^ range shows a clear difference between the pure PLCL sample and functionalized materials in the Amide I band region 1600–1700 cm^−1^ (Figure 10). Apart from the H1 sample that has a distinct peak with a maximum of around 1650 cm^−1^, all other samples show a significant elevation in that region above the result for a pure PLCL, however, the difference between each sample is negligible. For the H1 sample, a peak in the Amide II band region is also pronounced which further proves the presence of gelatin that is attached to the surface.

The results of BCA staining present gelatin amounts on the fibres’ surfaces at 0, 1, and 7 days of incubation in PBS and confirm what was observed in FTIR spectra analysis, that the highest gelatin amount was attached for the H1 sample (Figure 11). Much lower values were observed for other hydrolysis-activated samples with the tendency to decrease with a decline in the concentration of the solution. However, such a high value for the H1 sample could be the effect of not only the -COOH group concentration but also the high surface area of the fibre resulting from the surface fraying. There was also a positive correlation between the time of plasma treatment and the amount of gelatin. In the case of aminolysis, results are very similar for all samples without any relation between the amount of -NH_2_ groups and the amount of gelatin. The initial amount of the gelatin on the surface of reference samples, C and CEN, was relatively high, being as high as for the P10 and P40 samples, respectively.

For reference samples as well as for the H1 sample, a burst release of gelatin was observed after one day of incubation in PBS. That indicates a relatively high percentage of weakly bound absorbed gelatin, most probably due to physical interactions of polymer with gelatin molecules. In the case of the H1 sample, the likely cause of this result is its increased surface area associated with a strongly frayed surface of the fibres. For plasma-activated samples, this effect was much slighter, and even less visible for the other hydrolysed and aminolysed samples.

Results for the 7-day incubation were only slightly lower than for the 1-day incubation for all samples, which indicates the stability of gelatin coating. Except for H0.1, for all chemically modified samples, we observed higher values of gelatin amount than for both reference samples, with the highest amount equal to 9.61 ± 0.22 µg/mg for the P40 sample. It is worth noting that reference samples C and CEN with physically adsorbed gelatin provide 58% and 53% of the initial gelatin amount, respectively, after 7 days of incubation.

#### 3.2.2. Water Contact Angle

All samples demonstrated complete (0°) wettability after functionalization with gelatin. However, there were some differences observed in the time of water drop absorption into the scaffold depending on the type of modification. The changes in wettability were also observed for some samples after the incubation in PBS.

In the case of the reference samples (Figure 12a), the initial values of contact angle were equal to 92 ± 21.7° for the C sample, and 13 ± 29.3° for the CEN sample, and after 7 days of incubation, the initial contact angle was close to the angle before degradation for the C sample being equal to 80 ± 14.13° and increased to 58.65 ± 8.76° for the CEN sample. An increase was then observed in the absorption time after 7 days of incubation for the C sample, which can be related to the degradation of the coating.

Water drop on all hydrolysed samples before the degradation test was immediately absorbed, hence there is no dedicated graph. However, after incubation in PBS, the time of absorption slightly increased for hydrolysed samples (Figure 12b). There was no correlation between the amount of gelatin in these samples and the absorption time. At the 10 s time point, all hydrolysed samples have already absorbed the water drop.

In the case of aminolysed samples before the stability test, it was observed that the stronger the aminolysis parameters applied, the lower the initial contact angle value recorded. Interestingly, 7 day-incubation of PBS samples demonstrated higher wettability with immediate absorption of the water drop, which is in contrast to the reference and hydrolysed samples (Figure 12c).

#### 3.2.3. Mechanical Tests

The initial values of tensile strength and elongation at break for PLCL were equal to 7.0 ± 0.3 MPa and 427.6 ± 28.0%, respectively (Figure 13e,f). For both control samples, a significant decrease in tensile strength was observed after gelatin immobilisation (Figure 13a,e). This could be the effect of the applied temperature during gelatin immobilisation and the resulting change in the polymer structure. The plasma-treated samples demonstrated a slightly lower value of tensile strength in comparison to the reference samples (Figure 13b,e). In the case of the hydrolysed and aminolysed samples (Figure 13c–e), a gradual decrease in maximum load with the intensity of applied reaction conditions was observed. For the H1 sample, a dramatic decrease up to 3.2 ± 0.3 MPa was noticed. In the case of the samples after aminolysis, this is the effect of the change in the molecular weight. It was discussed in our previous study with reference to the theoretical explanation [61]. For hydrolysed samples, this could result from the surface defects generating stress in the material.

For the control sample with physically adsorbed gelatin, an increase in elongation at a break of 499.8 ± 9.1% was observed (Figure 13f). However, that was an unexpected result and not the case for the control sample activated with EDC/NHS. In the case of plasma-treated samples, there was no change in elongation at break. The hydrolysed samples showed a slight increase in the elongation at break for lower concentrations of NaOH, and a decrease for higher concentrations, especially for the H1 sample, which is a result of surface defects as in the case of tensile strength. For A1 and A2 aminolysed samples, only a slight decrease was detected. A more significant decrease was observed for the A3 sample; however, the maximum extension value was still high being equal to 372.8 ± 23.5%. Generally, a decrease in strain at break after aminolysis treatment is reported by other authors [62]. This is consistent with the results of the molecular weight change as a decrease in the M.W. results in a decrease in polymer chain entanglements and thus in the lower elongation at break.

#### 3.2.4. Cellular Studies

The results of in vitro cytotoxicity test performed on materials’ extracts showed some differences between the materials but did not indicate that any of the investigated samples were toxic to the cells (Figure 14a). All values were above 70% of the control. Plasma and NaOH treated materials (with the exception of the H1 sample) showed values above control which might mean some of the gelatin from the surface migrated to the medium and had a positive effect on the cells, but BCA test results do not prove this hypothesis.

The 0–3 day experiment where cells were cultured directly on the materials’ surfaces showed very similar and satisfactory results for all samples (Figure 14b). A slightly lower value was recorded for a control material with gelatin attached only via physisorption which is difficult to explain while a pure PLCL material without any attached gelatin obtained a result on par with the rest of the functionalized nonwovens.

The observations conducted via both scanning electron and fluorescence microscopy of fibroblast cells cultured on the materials’ surface showed some small differences in their morphology (Figure 15 and Figure 16). L929 fibroblasts are a type of adherent, connective cell line that in a living tissue are responsible for the production of an extracellular matrix and a fibrous environment is their natural habitat. The less favourable the surface of the material is, the rounder a fibroblast becomes in order to minimize its contact with it. In contrast, the more native-like the surface they are seeded on is, the more likely they are to spread, extending their filopodia and lamellipodia to attach themselves to the surface [63,64].

Tissue culture polystyrene/plastic (TCP) that multiwell plates are made of is hydrophilic, and fibroblasts demonstrate flattened, spear-like or triangular, elongated shapes when cultured on its surface. The cells cultured on the PLCL and C control samples, from which one was not functionalized at all, the second had gelatin-only physisorbed and showed inferior cell morphology compared to the CEN sample or TCP. There could be seen cells in round form, or if spread, showing a visibly small number of filopodia. In the case of the rest of functionalized materials, fibroblasts showed an improved morphology and spreading with large numerous filopodia and lamellipodia in comparison to the control materials without gelatin. Among all tested samples, the fibroblast cells cultured on samples H1 and P40 showed the greatest amount of filopodia indicating these materials possessed the most attractive surface properties to the L929 cells. In the case of the H1 sample, it is believed that such a result is a consequence of increased surface area. The surface roughness of the fibres caused by hydrolysis, as was first stated in the last paragraph, is favourable by fibroblasts as it mimics native ECM and provides more sites for attachment.

It is experimentally proven that surface chemistry modulates the conformation of adsorbed proteins [65,66]. Vallieres et al. [67] also reported differences in immobilised fibronectin conformation depending on the crosslinking method. In this study, as the -COOH groups of the polymer were activated during immersion in EDC/NHS solution after plasma and hydrolysis treatment, they were cross-linked with the -NH_2_ groups of gelatin. Similarly, in the case of aminolysis and GTA activation, -NH_2_ groups of gelatin were also involved in the cross-linking. Another thing to consider is if the -NH_2_ group, which is involved in the crosslinking, originates from the RGD sequence, it could affect its further biological activity. Currently, suitable peptide exposure after immobilisation is the topic of many studies [68,69,70]. The proposed aminolysis-based modification differs from the hydrolysis and plasma-based methods, as in that case there is a spacer between the material surface and protein, consisting of diamine and glutaraldehyde chains. Other authors reported that this could improve the biological efficiency of the biomolecule [71,72]. However, such a positive effect of the spacer presence was not observed in this study.

## 4. Conclusions

The main goal of this work was to compare three surface activation methods—hydrolysis, aminolysis, and cold oxygen plasma in terms of both their effect on PLCL material, as well as on enabling successful gelatin immobilization. The first part of the work consisted of the optimisation of the conditions of those processes. Discarding all samples with significantly altered morphology of the fibres and choosing the group with hydrophilic ones enabled the main part of the work to be conducted on a set of materials with more uniform properties, which facilitated the comparison of the final results of functionalized materials.

The results of all of the experiments indicated that all investigated surface activation methods were successful in being the first step of the gelatin immobilisation process. A highly hydrophilic material with attached gelatin, which is more stable compared to the control samples, was achieved for all three tested methods.

Cellular studies showed improved morphology and spreading of cells cultured on functionalized materials compared to the untreated PLCL or the sample with physisorbed gelatin only.

A comprehensive examination of functionalized materials revealed differences in their properties that have to be taken into consideration when selecting which one of them is the most suitable for a specific medical application. A decision should be based on a specific function that the final scaffold material has to perform and the properties it needs to possess.

The most noticeable differences between the three surface activation methods were in how they affected the molecular weight of the polymer. Although both cold oxygen plasma and aminolysis caused polymer chain degradation in bulk, the hydrolysis induced damage on the surface of the material, whereas fibres’ cores were barely affected.

The frayed surface of the fibres observed for samples hydrolysed with 1 M NaOH did help with gelatin attachment, as shown in the BCA test, but such harsh treatment resulted in the lack of mechanical properties and the loss of the almost 60% of material mass. Too many defects introduced to fibres’ surfaces through hydrolysis weakened the material, making it prone to tearing. Aminolysis and plasma treatment with the loss of molecular weight increased brittleness.

These observations indicate that the use of any surface activation method should be limited to the lowest concentration/reaction time that enables subsequent satisfactory functionalization.

Future work following these findings should be focused on the optimisation of a surface activation method and subsequent attachment of a biologically active molecule for a specific application. Further biological studies conducted on functionalised materials must be performed with the use of cell types relevant to the chosen application.

## Figures and Tables

**Figure 1 jfb-13-00272-f001:**
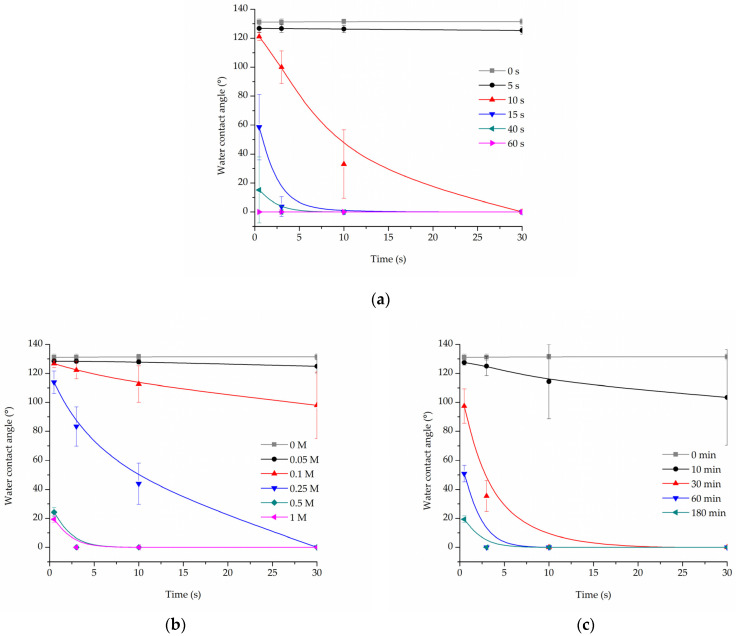
Water contact angle of PLCL nonwoven samples after surface activation with (**a**) oxygen plasma, (**b**) NaOH solutions with increasing concentrations for 3 h, (**c**) 1 M NaOH solution for a set of time periods up to 180 min.

**Figure 2 jfb-13-00272-f002:**
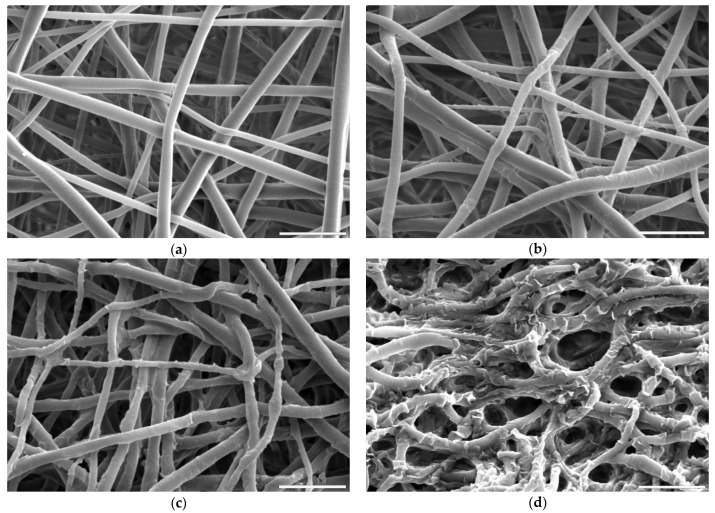
Scanning electron microscope images of PLCL nonwoven sample (**a**) untreated, after surface activation with (**b**) 0.5 M NaOH for 3 h, (**c**) 1 M NaOH for 3 h, (**d**) 1 M NaOH for 6 h. The marker is equal to 5 μm.

**Figure 3 jfb-13-00272-f003:**
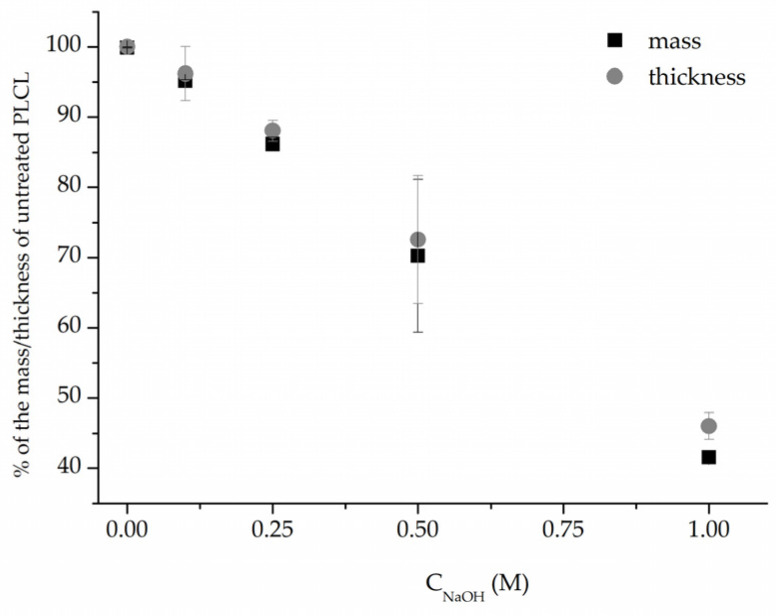
Mass and thickness change after 3 h NaOH surface activation of PLCL nonwoven.

**Figure 4 jfb-13-00272-f004:**
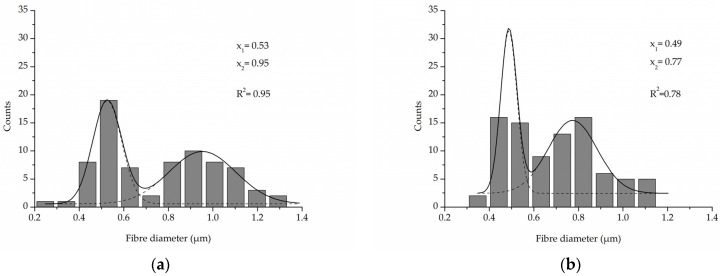
Fibre diameter distribution with local maxima and regression values of (**a**) an untreated PLCL sample, (**b**) a sample treated with 1 M NaOH for 3 h. Data were fitted with Gaussian function.

**Figure 5 jfb-13-00272-f005:**
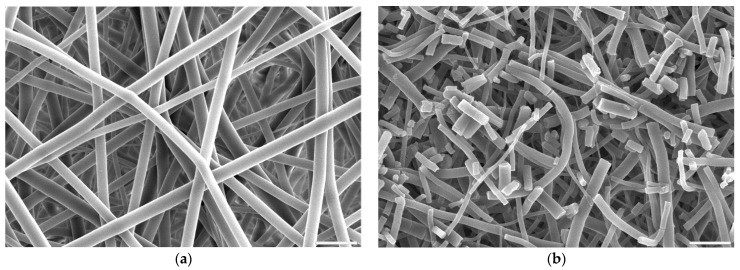
Scanning electron microscope images of PLCL nonwoven samples after surface activation at 30 °C with (**a**) 6% diamine for 5 min, (**b**) 10% diamine for 30 min. The marker is equal to 5 μm.

**Figure 6 jfb-13-00272-f006:**
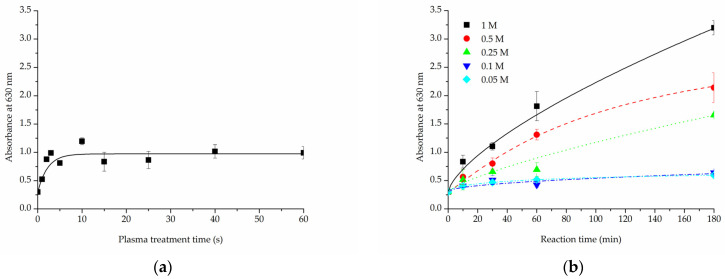
Toluidine Blue O test results for PLCL samples treated with (**a**) cold oxygen plasma, (**b**) NaOH with varying concentrations and reaction times.

**Figure 7 jfb-13-00272-f007:**
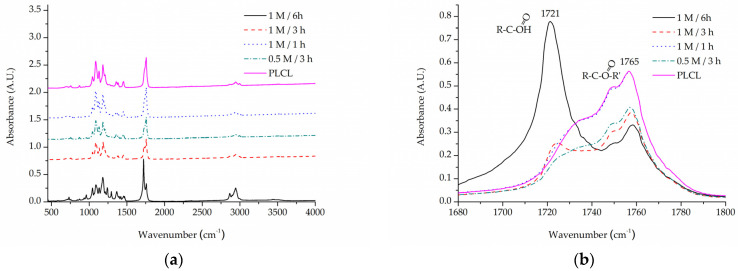
A comparison of IR spectra for PLCL samples after hydrolysis, (**a**) full spectrum, (**b**) 1680–1800 cm^−1^ range that shows absorption of C=O bond stretching frequency.

**Figure 8 jfb-13-00272-f008:**
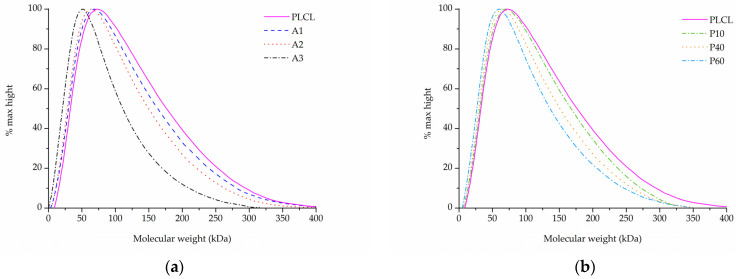
Molecular weight of PLCL treated with (**a**) aminolysis, (**b**) oxygen plasma.

**Figure 9 jfb-13-00272-f009:**
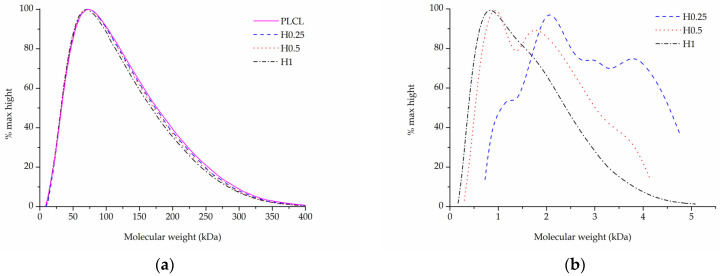
Molecular weight of PLCL treated with NaOH, (**a**) main peak, (**b**) small molecular weight peak.

**Figure 10 jfb-13-00272-f010:**
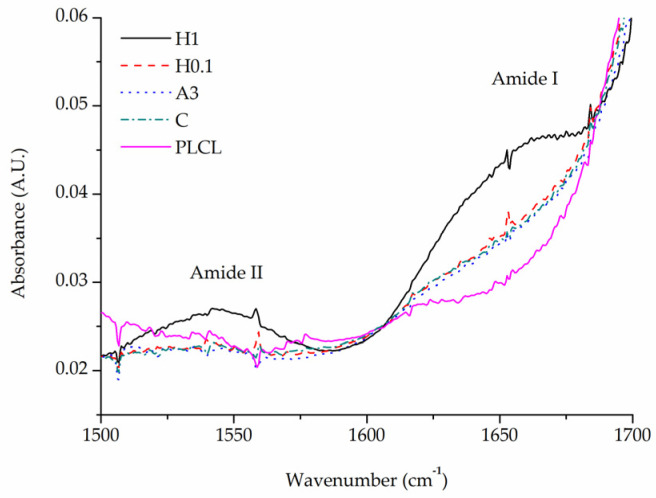
Comparison of FTIR spectra of functionalized samples.

**Figure 11 jfb-13-00272-f011:**
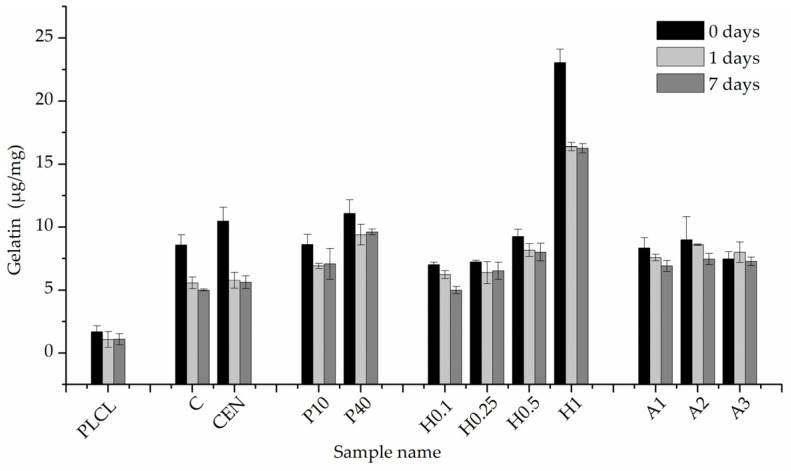
Gelatin amount on the material’s surface after 0, 1 day, and 7 days of incubation in PBS.

**Figure 12 jfb-13-00272-f012:**
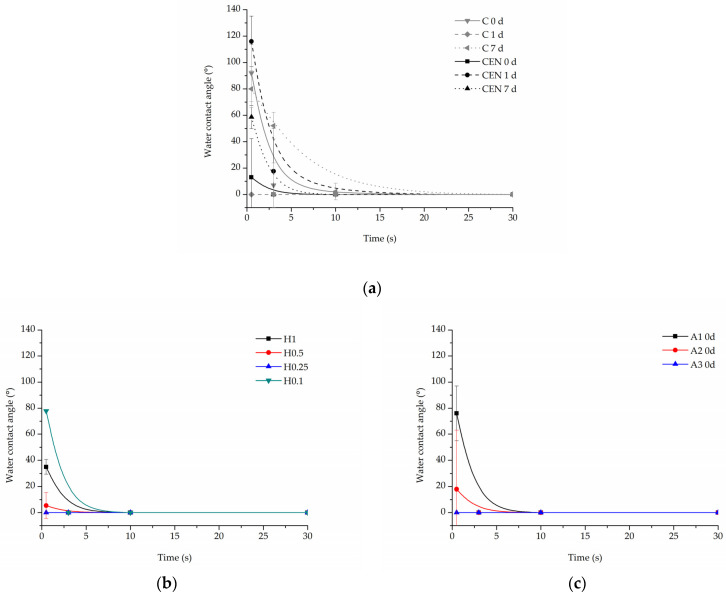
Water contact angle of PLCL nonwoven samples after surface functionalization: (**a**) reference samples after 0, 1, and 7 days of incubation in PBS, (**b**) hydrolysed samples after 7 days of incubation in PBS, and (**c**) aminolysed samples after surface functionalization before incubation in PBS.

**Figure 13 jfb-13-00272-f013:**
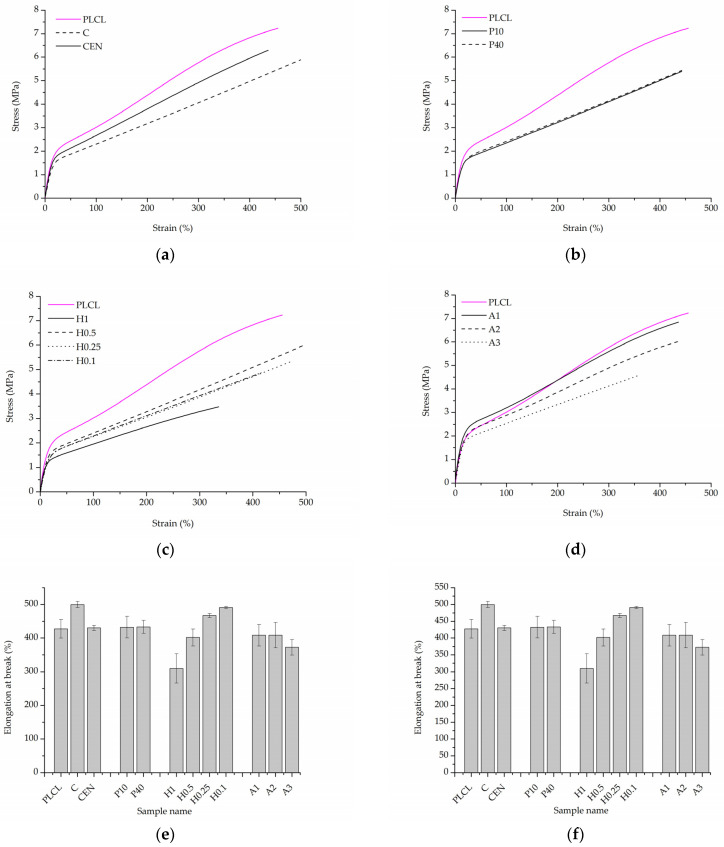
Uniaxial tensile stress test results. Stress–strain curves for: (**a**) control samples, (**b**) materials treated with plasma, (**c**) hydrolysis, and (**d**) aminolysis. Graphs comparing: (**e**) tensile strength, (**f**) elongation at break of investigated samples.

**Figure 14 jfb-13-00272-f014:**
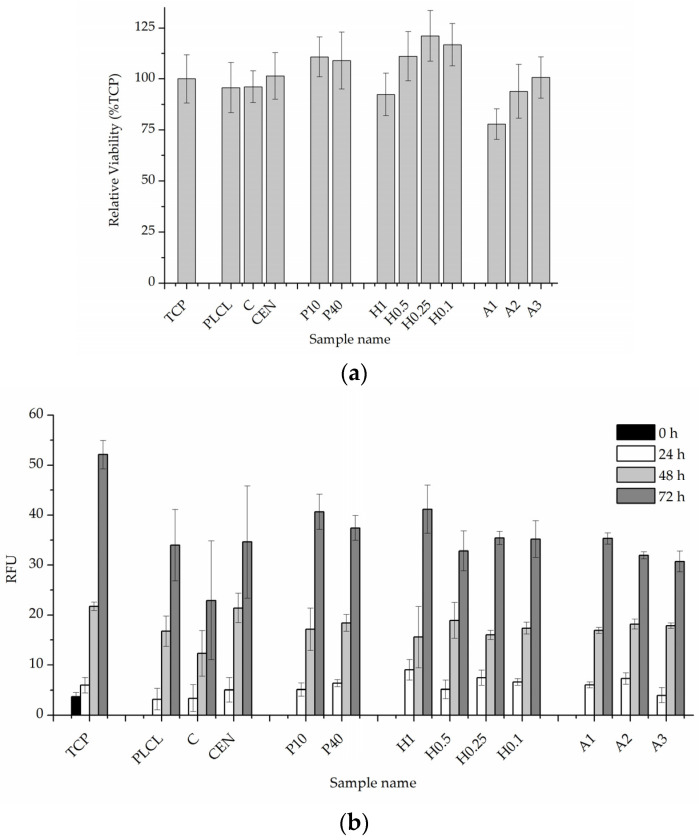
Quantitative in vitro test results, (**a**) 24-h cytotoxicity on material extracts, (**b**) 0–3 day viability, and proliferation test in direct contact with the material.

**Figure 15 jfb-13-00272-f015:**
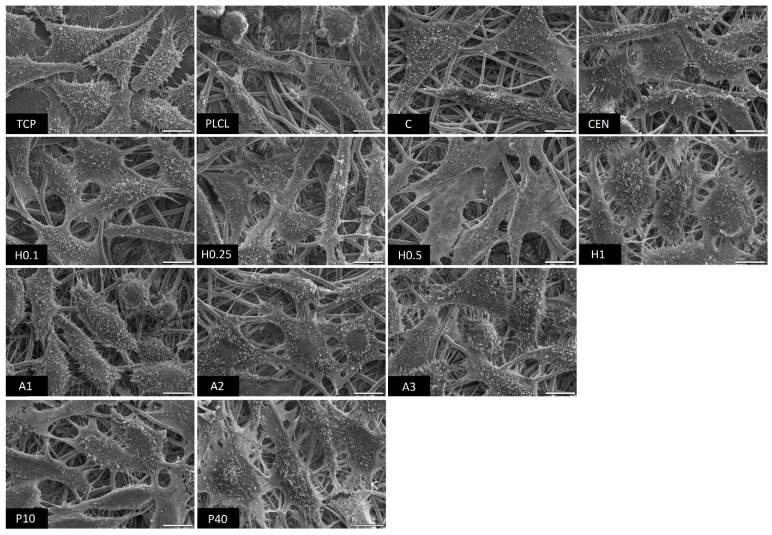
SEM microscopy images of L929 cells cultured the functionalized nonwoven materials and TCP. The marker is equal to 5 μm.

**Figure 16 jfb-13-00272-f016:**
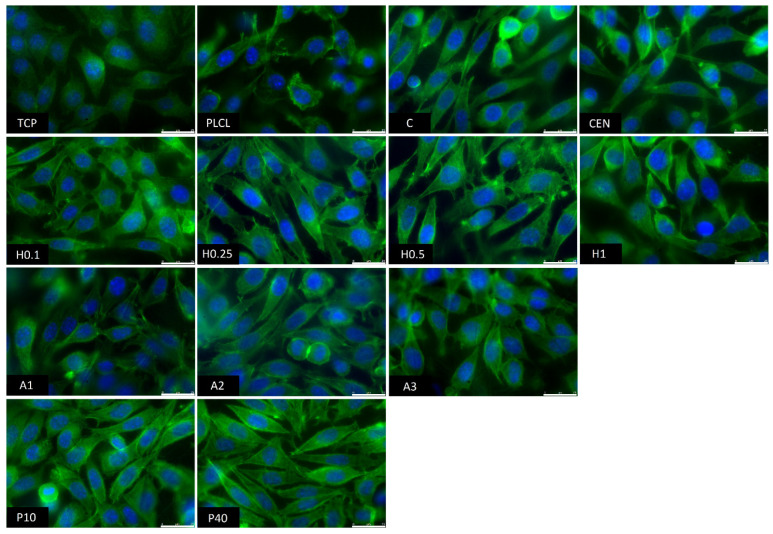
Fluorescence microscopy images of L929 cells cultured the functionalized nonwoven materials and TCP. The marker is equal to 25 μm.

**Table 1 jfb-13-00272-t001:** Initial set of surface activation conditions applied in the optimisation step.

Activation Method	Optimisation Conditions	
	Treatment Time	Reagent Concentration
Cold oxygen plasma	5, 10, 15, 40, 60 (s)	-
Alkaline hydrolysis	10, 30, 60, 180 (min)	0.05, 0.1, 0.25, 0.5, 1 (M NaOH)
Aminolysis	5, 10, 15, 30 (min)	2, 6, 10 (% *w*/*v* EDA in isopropanol)

**Table 2 jfb-13-00272-t002:** Names of all samples that underwent functionalization.

Surface Activation Method	Time	Reagent Concentration	Crosslinking Method	Name
Hydrolysis	3 h	1 M	EDC/NHS	H1
		0.5 M		H0.5
		0.25 M		H0.25
		0.1 M		H0.1
Plasma	10 s	-	EDC/NHS	P10
	40 s			P40
Aminolysis	5 min	2% *w*/*v*	GTA	A1
	10 min	2% *w*/*v*	GTA	A2
	5 min	6% *w*/*v*	GTA	A3
Control (no activation)	-	-	EDC/NHS	CEN
			-	C

**Table 3 jfb-13-00272-t003:** The changes in weight (M_w_) and number (M_n_) average molecular weight of PLCL after surface activation with different methods. Data for samples A1–A3 come from previously published work [50].

Sample	M_n_ (kDa)	M_w_ (kDa)	% M_nPLCL_	% M_wPLCL_	PDI
PLCL	58.7	88.3	100%	100%	1.50
A1	49.8	81.3	85%	92%	1.63
A2	47.8	75.9	81%	86%	1.59
A3	30.6	58.4	52%	66%	1.91
P10	54.5	82.1	93%	93%	1.51
P40	52.5	77.5	89%	88%	1.48
P60	46.0	71.3	78%	81%	1.55
H1	57.7	85.3	98%	97%	1.48
H0.5	57.3	85.9	97%	97%	1.50
H0.25	59.6	87.7	100%	99%	1.47
H0.1	56.4	85.2	96%	97%	1.51
H0.05	57.3	86.2	97%	98%	1.51

## Data Availability

The data presented in this study are available on request from the corresponding author.

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
