# Peer review of "A Comparative Study of Three Approaches to Fibre’s Surface Functionalization"

_jfb, 2022, doi:10.3390/jfb13040272_

Round 1

Reviewer 1 Report

In the manuscript “A comparative study of three approaches to fibre’s surface functionalization”, three approaches were employed to functionalize the surface of the poly(lactide-co-ε-caprolactone) PLCL fibres with gelatin. It was demonstrated that hydrolysis influences the surface of the material the most, while plasma treatment and aminolysis have an effect on the whole volume of the material. The work is well designed, however, the authors should pay more attention on correctness of the experimental procedures and results. Some issues need to be addressed before it can be accepted.   

1. All images in the manuscript should be vertically aligned to the middle with the captions.

2.  The time coordinate in the diagrams should start about 0 and end in predetermined time in Figure 1 and Figure 6.

3. Please change the square brackets to parenthese in vertical and abscissa unit of diagrams.
4.  In Figure 8 and Figure 9, please uniform the vertical and abscissa caption in full name instead of abbreviated form.

5.  In Figure 11, Figure 13 and Figure 14, please complement the abscissa caption in full name.

6. Please revise the manuscript to improve grammar and English language usage throughout, potentially with the assistance of an experienced scientific writer or editor.
7. A lot of work has been done to demonstrate that the hydrolysis influences the surface of the material the most, while plasma treatment and aminolysis have an effect on the whole volume of the material. Thus, please add the function mechanism of different chemical process regarding to  L929 cells response in the Conclusion Part.

Author Response

In the manuscript “A comparative study of three approaches to fibre’s surface functionalization”, three approaches were employed to functionalize the surface of the poly(lactide-co-ε-caprolactone) PLCL fibres with gelatin. It was demonstrated that hydrolysis influences the surface of the material the most, while plasma treatment and aminolysis have an effect on the whole volume of the material. The work is well designed, however, the authors should pay more attention on correctness of the experimental procedures and results. Some issues need to be addressed before it can be accepted.   

  1. All images in the manuscript should be vertically aligned to the middle with the captions.
  • All formatting, the way all of the images and captions are placed in the manuscript was done accordingly to Journal’s requirements. All paragraph as well as captions are to be placed to the left side of a page.
  1.  The time coordinate in the diagrams should start about 0 and end in predetermined time in Figure 1 and Figure 6.
  • Scale on the x-axis has been corrected as suggested.
  1. Please change the square brackets to parenthese in vertical and abscissa unit of diagrams.
  • The square brackets to have been changed to parenthese in all figures.

  1.  In Figure 8 and Figure 9, please uniform the vertical and abscissa caption in full name instead of abbreviated form.
  • Corrected as suggested.
  1.  In Figure 11, Figure 13 and Figure 14, please complement the abscissa caption in full name.
  • Labels have been added as suggested.
  1. Please revise the manuscript to improve grammar and English language usage throughout, potentially with the assistance of an experienced scientific writer or editor.
  • We scanned the manuscript for errors and corrected as many as we were able to find. The manuscript has been reviewed by an English language specialist.

  1. A lot of work has been done to demonstrate that the hydrolysis influences the surface of the material the most, while plasma treatment and aminolysis have an effect on the whole volume of the material. Thus, please add the function mechanism of different chemical process regarding to  L929 cells response in the Conclusion Part
  • All three surface activation methods together with complementary gelatin attachment reaction bind gelatin molecules to the surface covalently. Depending on crosslinking agent used, GTA, or EDC/NHS, the mechanism of binding is different, but the question of how these differences affects cell response is a very complex matter. It is discussed in the last paragraph before Conclusions. If you comment referred more to the question of how an activation method influenced the material: more on the surface or in bulk – an explanation of how fried, uneven surface caused by hydrolysis affected cell response was added in ‘Cellular studies’.

Reviewer 2 Report

The paper compares three different approaches for the functionalization of PLCL fiber's surface in scaffold applications. As a first suggestion the title should be include the application sector of the studed fibers. In the introduction the acronymous ECM is reported the first time without quoting the individual words in full. In the materials and methods section there are some missing informations:

1) in plasma treatment is not specified the distance betwwn the treated surface and the plasma gun.

2) in mechanical tests it is not specified the thickness and the method of measuring elongation

In results:

3) FIgure 1 should better reports the differences (plasma etc..)

4) FIgure 2 the numbers on the scales are not readable.

The study is very interesting, the results adequately presented and after a minor revision could be published.

Author Response

The paper compares three different approaches for the functionalization of PLCL fiber's surface in scaffold applications. As a first suggestion the title should be include the application sector of the studed fibers. In the introduction the acronymous ECM is reported the first time without quoting the individual words in full. In the materials and methods section there are some missing informations:

  • Concerning the title: In our work the main objective was to compare these three surface activation methods in terms of their effect on PLCL material without a specific tissue engineering application in mind. We aimed at providing comprehensive data that could be useful to many people while deciding what method to use for their specific needs. Because of that we are not including/specifying any application in the title of this work.
  • The ‘extracellular matrix (ECM)’ is now stated in full where it appears the first time in the manuscript.

1) in plasma treatment is not specified the distance between the treated surface and the plasma gun.

  • The Plasma surface treatment machine Zepto (Diener, Germany) that was used here does not have a plasma gun. Instead, a sample is placed in a chamber on a glass shelf and is treated from each side, on each surface at once. See the attached photograph of the device.

2) in mechanical tests it is not specified the thickness and the method of measuring elongation

  • The thickness of each sample (50 mm x 5 mm) was measured on both ends and those values were averaged. This information can be found in the manuscript. All samples’ thickness varied between 90 µm and 135 µm. Thickness values were not included in the manuscript, since they are only needed to calculate stress values.

  • The elongation value in mm is provided by the machine used to perform mechanical tests and elongation in % is then calculated from the known sample length that is used in the test – here 20 mm.

In results:

3) FIgure 1 should better reports the differences (plasma etc..)

  • The data presented in each graph that Figure 1 consists of, is described in separate paragraph in section ‘3.1.1. Water contact angle’. Figure 1a-c show how the time of treatment / reagent concentration of each method affects the hydrophilicity of PLCL fibres. Each of the paragraphs puts in focus the most prominent points where hydrophilicity changes within the range of a studied parameter.

4) FIgure 2 the numbers on the scales are not readable.

  • There are no numbers next to the scale bar in all SEM images. In both Figure 2 and 5, the length of a scale bar is defined in the caption.

Author Response

Answer 1

To all comments concerning specification of ASTM used in this work – We did not perform the experiments discussed in this paper in accordance with any ASTM, since in our laboratory we do not have any ASTM implemented.

Answer 2

To all comments “Please unbold (a), (b), (c) etc in the figure and caption” -  all letters in bold in figures and  captions were formatted this way because of Journal’s requirements. All editing was done following  Journal’s official template.

Answer 3

The quality of all figures has been improved. The font type has been changed to Palatino Linotype. The font size in all wordings and values has been increased by 4 to 8 points. Labels were added where missing.

Answer 4

This work consists of both optimization of surface activation methods as well as a direct comparison of effects that these methods have on poly(lactide-co-ε-caprolactone) fibres. In order to compare these findings with previously published work, those results would have to concern the same type of material, at least similar reaction conditions, as well as methods used to asses them. Such experimental work has not been published so far, hence it is not possible to draw any reliable comparison this way. While there is literature concerning methods used here separately, they are performed on differently formulated materials, different polymers and using reagent concentration/reaction times that barely overlap with those used in this studies. We believe that the novelty of our work can be greatly attributed to this very fact that not much similar studies has been performed and published so far.

ABSTRACT:

  1. Abstract need to restructure. Should start with Introduction, problem, research gap/motivation, objective, methodology, results and significant of study
  • Abstract has been improved as suggested.
  1. Please include results obtained in abstract.
  • The results and findings appear in the abstract starting from ‘It was shown…’, till the end of it.
  1. Page 1, Line 10: ‘-COOH’ and ‘-COOH and -NH2’ should write full name and then (-COOH) and (-NH2) in a bracket. Please take note, for 1st time appear in manuscript, should write full name and then write abbreviation/symbol in a bracket. Please check the whole manuscript and do correction accordingly.
  • Corrected as suggested

KEYWORDS: Please replace coma symbol (,) with semi colon symbol (;) between keyword

  • Corrected as suggested

INTRODUCTION:

  1. Page 1, Line 32: Please avoid use word they, we, their etc. Please check the whole manuscript and do correction accordingly.
  • Changed to ‘Polymers’
  1. Page 1, Lines 38-39: Please add citation/s for this statement ‘However, the … lack of bioactivity [X].’ Please take note, all fact statements need to have citation/s. Please check the whole manuscript and do correction accordingly.
  • A citation has been added.
  1. Page 1, Line 41: ‘ECM’ should write full name and then (ECM) in a bracket. Please take note, for 1st time appear in manuscript, should write full name and then write abbreviation/symbol in a bracket. Please check the whole manuscript and do correction accordingly.
  • Corrected as suggested
  1. Page 2, Line 47: Just write ECM because of 2nd time appear in this manuscript. Please check the whole manuscript and do correction accordingly.
  • Corrected as suggested
  1. Page 2, Lines 61-62: Please add citation/s for this statement ‘Another physical …. materials [X].’ Please take note, all fact statements need to have citation/s. Please check the whole manuscript and do correction accordingly.
  • A citation has been added.
  1. Page 2, Line 94: Please write, ‘Therefore, the main objectives…’
  • Corrected as suggested
  1. Pages 2-3, Lines 94-106: Please combine these 2 paragraphs and make it 1 paragraph.
  • Corrected as suggested. Please notice, combined paragraphs appear as such only after changes are accepted.
  1. Page 3, Line 101: ‘EDC/NHS’ should write full name and then (EDC/NHS) in a bracket. Please take note, for 1st time appear in manuscript, should write full name and then write abbreviation/symbol in a bracket. Please check the whole manuscript and do correction accordingly.
  • Corrected as suggested
  1. Page 3, Line 114: ‘BCA’ should write full name and then (BCA) in a bracket. Please take note, for 1st time appear in manuscript, should write full name and then write abbreviation/symbol in a bracket. Please check the whole manuscript and do correction accordingly.
  • Corrected as suggested
  1. Page 3, Line 142: Please avoid use word they, we, their etc. Please check the whole manuscript and do correction accordingly.
  • In the sentence ‘All samples were then subjected to a series of tests to assess their morphology, physicochemical properties and activation efficiency.’ the use of ‘their’ is justified, do not introduce any ambiguity and it is not used incorrectly.

MATERIALS AND METHODS

  1. Page 4, Table 1: Please use bracket ( ) and not [ ]
  • corrected
  1. Page 4, Line 153: ‘NaOH’ should write full name and then (NaOH) in a bracket. Please take note, for 1st time appear in manuscript, should write full name and then write abbreviation/symbol in a bracket. Please check the whole manuscript and do correction accordingly.
  • ‘sodium hydroxide (NaOH)’ was added in section 2.1, where it was missing.
  1. Page 4, Line 166: Please specify ASTM used for samples preparation
  • See the answer 1 at the top.
  1. Page 4, Line 178: Please write as 1.42±0.08 x 10-8 mol/mg, 5.43±0.90 x 10-8 mol/mg, and 8.63±1.16 x 10-8 mol/mg
  • Corrected as suggested.
  1. Page 5, Line 194: Please specify ASTM used for samples preparation
  • See the answer 1 at the top.
  1. Page 6, Line 216: Please state manufacturer and country for JSM-6010PLUS/LV InTouchScope™
  • Corrected as suggested.
  1. Page 6, Line 223: Please avoid use word they, we, their etc. Please check the whole manuscript and do correction accordingly.
  • Corrected as suggested.
  1. Page 6, Line 227: Please state manufacturer and country for UV-vis spectrophotometer
  • Corrected as suggested.
  1. Page 6, Line 230: ‘FTIR’ should write full name and then (FTIR) in a bracket. Please take note, for 1st time appear in manuscript, should write full name and then write abbreviation/symbol in a bracket. Please check the whole manuscript and do correction accordingly.
  • Corrected as suggested.
  1. Page 6, Line 231: ‘ATR’ should write full name and then (ATR) in a bracket. Please take note, for 1st time appear in manuscript, should write full name and then write abbreviation/symbol in a bracket. Please check the whole manuscript and do correction accordingly.
  • Corrected as suggested.
  1. Page 6, Line 239: Please state manufacturer and country for Shimadzu Nexera device
  • Corrected as suggested.
  1. Page 6, Line 240: ‘TGF’ should write full name and then (TGF) in a bracket. Please take note, for 1st time appear in manuscript, should write full name and then write abbreviation/symbol in a bracket. Please check the whole manuscript and do correction accordingly.
  • THF, an eluent used in chromatography, in full name ‘tetrahydrofuran’, appears for the first time in section 2.1
  1. Page 6, Line 240: ‘GPC’ should write full name and then (GPC) in a bracket. Please take note, for 1st time appear in manuscript, should write full name and then write abbreviation/symbol in a bracket. Please check the whole manuscript and do correction accordingly.
  • Corrected as suggested.
  1. Page 6, Line 243: ‘PTFE’ should write full name and then (PTFE) in a bracket. Please take note, for 1st time appear in manuscript, should write full name and then write abbreviation/symbol in a bracket. Please check the whole manuscript and do correction accordingly.
  • Corrected as suggested.
  1. Page 6, Line 247: Please write as ‘…where α is 0.7, K is 14.1 x 10-5 dl/g…’
  • Corrected as suggested.
  1. Page 6, Line 247: Please write as ‘…α is 0.68, K is 4.84 x 10-4 dl/g…’
  • Corrected as suggested.
  1. Page 6, Lines 248-249: Please write as ‘… Nuuttilla M. et al. [51] were used.’
  • Corrected as suggested.
  1. Page 6, Line 252: Please specify ASTM used for Uniaxial tensile testing samples preparation
  • See the answer 1 at the top.
  1. Page 7, Line 269: Please state brand, manufacturer and country for QuantiPro™
  • The brand name of BCA kit is QuantiPro™. Both manufacturer and country are stated in section 2.1.
  1. Page 7, Line 279: Please state brand, manufacturer and country for cellular tests
  • ‘cellular tests’ phrase here refers to a group of experiments where cells are used, explained in detail in sections 2.6.1. and 2.6.2.
  1. Page 7, Line 287: Please specify ASTM used for cytotoxicity test
  • See the answer 1 at the top.
  1. Page 7, Line 301: ‘PBS’ should write full name and then (PBS) in a bracket. Please take note, for 1st time appear in manuscript, should write full name and then write abbreviation/symbol in a bracket. Please check the whole manuscript and do correction accordingly.
  • PBS in full name ‘phosphate buffer saline’ appears for the first time in section 2.1
  1. Page 7, Line 309: ‘SEM’ should write full name and then (SEM) in a bracket. Please take note, for 1st time appear in manuscript, should write full name and then write abbreviation/symbol in a bracket. Please check the whole manuscript and do correction accordingly.
  • Corrected as suggested.

RESULTS AND DISCUSSION

  1. Page 8, Figure 1: Please improve quality of all figures. Make sure all lines, values and wording are readable. Please use standard size and font type for all wordings and values at all figure. Suggest to use Palatino Linotype (type font), text size 9. Please check the whole manuscript and do correction accordingly.
  • See the answer 3 at the top.
  1. Page 8, Figure 1: Please unbold (a), (b), (c) etc in the figure and caption
  • See the answer 2 at the top.
  1. Page 9, Figure 2: Please add labels on figures to indicate what actually authors want to show
  • A detailed description of images in Figure 2 can be found in the caption underneath.
  1. Page 9, Figure 2: Please unbold (a), (b), (c) etc in the figure and caption
  • See the answer 2 at the top.
  1. Page 10, Figures 3 and 4: Please improve quality of all figures. Make sure all lines, values and wording are readable. Please use standard size and font type for all wordings and values at all figure. Suggest to use Palatino Linotype (type font), text size 9. Please check the whole manuscript and do correction accordingly.
  • See the answer 3 at the top.
  1. Page 10, Figures 3 and 4: Please unbold (a), (b), (c) etc in the figure and caption
  • See the answer 2 at the top.
  1. Page 11, Lines 383-389: Please compare your findings with other previous researchers and not only your previous researchers. Is it inline? Contradict? Please discuss critically
  • A commentary in reference to previous research concerning this matter has been added together with a citation. Additionally see the answer 4 at the top.
  1. Page 11, Figure 5: Please add labels on figures to indicate what actually authors want to show
  • A detailed description of images in Figure 5 can be found in the caption underneath.
  1. Page 11, Figure 5: Please unbold (a), (b) etc in the figure and caption
  • See the answer 2 at the top.
  1. Page 11, Line 393, section 3.1.3: Please compare results/findings obtained in this section with previous researchers. Is it inline? Contradict? Please discuss critically
  • See the answer 4 at the top.
  1. Page 11, Figure 6: Please improve quality of all figures. Make sure all lines, values and wording are readable. Please use standard size and font type for all wordings and values at all figure. Suggest to use Palatino Linotype (type font), text size 9. Please check the whole manuscript and do correction accordingly.
  • See the answer 3 at the top.
  1. Page 11, Figure 6: Please unbold (a), (b) etc in the figure and caption
  • See the answer 2 at the top.
  1. Page 12, Figure 7: Please unbold (a), (b) etc in the figure and caption
  • See the answer 2 at the top.
  1. Page 12, Lines 429-431: Please compare this statement with previous researchers. Is it inline? Contradict? Please discuss critically
  • See the answer 4 at the top.
  1. Page 12, Line 433, section 3.1.4: Please compare results/findings obtained in this section with other previous researchers and not only your previous findings. Is it inline? Contradict? Please discuss critically
  • A commentary in reference to previous research concerning this matter has been added together with a citation. Additionally see the answer 4 at the top.
  1. Page 13, Line 446: Please write as ‘…activation with different methods [49].’
  2. Page 13, Line 447: Please remove or if want, can mention in text ‘*Data from previously published work [49].’
  • Answer to comments 46 and 47: The footer with the citation is now incorporated into the Table 3 name.
  1. Page 13, Figure 8: Please unbold (a), (b) etc in the figure and caption
  • See the answer 2 at the top.
  1. Page 14, Figure 9: Please unbold (a), (b) etc in the figure and caption
  • See the answer 2 at the top.
  1. Page 14, Line 459: ‘Mw and Mn’ should write full name and then (Mw) and (Mn) in a brackets. Please take note, for 1st time appear in manuscript, should write full name and then write abbreviation/symbol in a bracket. Please check the whole manuscript and do correction accordingly.
  • ‘(Mw)’ and ‘(Mn)’ are now added in the Table 3 description, where ‘weight and number averaged molecular weight’ appear for the first time in the manuscript.
  1. Page 14, Line 477, section 3.2.1: Please compare results/findings obtained in this section with other previous researchers and not only your previous findings. Is it inline? Contradict? Please discuss critically
  • See the answer 4 at the top.
  1. Page 14, Line 479: ‘BCA’ should write full name and then (BCA) in a bracket. Please take note, for 1st time appear in manuscript, should write full name and then write abbreviation/symbol in a bracket. Please check the whole manuscript and do correction accordingly.
  • ‘bicinchoninic acid’ which is a full name of ‘BCA’ was added in section 2.1 where all used materials are listed.
  1. Page 14, Line 485: Please avoid use word they, we, their etc. Please check the whole manuscript and do correction accordingly.
  • Corrected as suggested.
  1. Page 15, Figure 10: Please improve quality of all figures. Make sure all lines, values and wording are readable. Please use standard size and font type for all wordings and values at all figure. Suggest to use Palatino Linotype (type font), text size 9. Please check the whole manuscript and do correction accordingly.
  • See the answer 3 at the top.
  1. Page 15, Figure 11: Please improve quality of all figures. Make sure all lines, values and wording are readable. Please use standard size and font type for all wordings and values at all figure. Suggest to use Palatino Linotype (type font), text size 9. Please check the whole manuscript and do correction accordingly.
  • See the answer 3 at the top.
  1. Page 15, Line 530: Please avoid use word they, we, their etc. Please check the whole manuscript and do correction accordingly.
  • Corrected as suggested.
  1. Page 15, Line 538: Please avoid use word they, we, their etc. Please check the whole manuscript and do correction accordingly.
  • Corrected as suggested.
  1. Pages 16-17, Figure 12: Please improve quality of all figures. Make sure all lines, values and wording are readable. Please use standard size and font type for all wordings and values at all figure. Suggest to use Palatino Linotype (type font), text size 9. Please check the whole manuscript and do correction accordingly.
  • See the answer 3 at the top.
  1. Pages 16-17, Figure 12: Please unbold (a), (b) etc in the figure and caption
  • See the answer 2 at the top.
  1. Page 17, Lines 549-551: Please remove sentence ‘This section may … can be drawn.’
  • Corrected as suggested.
  1. Page 17, Lines 552-564: Please compare results/findings obtained in this section with other previous researchers and not only your previous findings. Is it inline? Contradict? Please discuss critically
  • See the answer 4 at the top.
  1. Page 17, Lines 571-572: ‘For A1 and A2 aminolysed samples only a slight decrease was detected.’ Please compare results/findings reported in this statement with other previous researchers and not only your previous findings. Is it inline? Contradict? Please discuss critically
  • A commentary in reference to previous research concerning this matter has been added together with a citation. Additionally see the answer 4 at the top.
  1. Page 17, Line 575: ‘M.W.’ should write full name and then (M.W.) in a bracket. Please take note, for 1st time appear in manuscript, should write full name and then write abbreviation/symbol in a bracket. Please check the whole manuscript and do correction accordingly.
  • See the answer to the comment no. 50.
  1. Page 18, Figure 13: Please improve quality of all figures. Make sure all lines, values and wording are readable. Please use standard size and font type for all wordings and values at all figure. Suggest to use Palatino Linotype (type font), text size 9. Please check the whole manuscript and do correction accordingly.
  • See the answer 3 at the top.
  1. Page 18, Figure 13: Please unbold (a), (b) etc in the figure and caption
  • See the answer 2 at the top.
  1. Page 18, Line 583, section 3.2.4: Please compare results/findings obtained in this section with other previous researchers and not only your previous findings. Is it inline? Contradict? Please discuss critically
  • See the answer 4 at the top.
  1. Page 19, Figure 14: Please improve quality of all figures. Make sure all lines, values and wording are readable. Please use standard size and font type for all wordings and values at all figure. Suggest to use Palatino Linotype (type font), text size 9. Please check the whole manuscript and do correction accordingly.
  • See the answer 3 at the top.
  1. Page 19, Figure 14: Please unbold (a), (b) etc in the figure and caption
  • See the answer 2 at the top.
  1. Page 19, Lines 600-603: Please avoid use word they, we, their etc. Please check the whole manuscript and do correction accordingly.
  • In the sentence ‘L929 fibroblasts are a type of adherent, connective cell line that in a living tissue are responsible for production of extracellular matrix and a fibrous environment is their natural habitat.’ the use of ‘their’ is justified and it is not used incorrectly.
  1. Page 20, Figures 15 and 16: Please add labels on figures to indicate what actually authors want to show
  • All labels with a sample’s name are placed in down right corner of each image.
  1. Page 21, Line 627: Please write as ‘Vallieres et al. [62]..’ and please remove citation at the end of sentence.
  • Corrected as suggested.
  1. Page 21, Lines 630-639: Please avoid use word they, we, their etc. Please check the whole manuscript and do correction accordingly.
  • Line 639: Corrected.
  • In the sentence ‘…as the ­COOH groups of the polymer were activated during immersion in EDC/NHS solution after plasma and hydrolysis treatment, they were crosslinked with the ­NH2 groups of gelatin.’ the use of ‘they’ is justified and it is not used incorrectly.

CONCLUSIONS:

  1. Please write the conclusion in a point form.
  • We believe that presenting each important conclusion in a separate paragraph, instead in a point form is commonly used and correct. While using points to list conclusions formulated in short, clipped sentences might be optimal, presenting ‘Conclusion’ section of this manuscript in such a way in not beneficial to their perception.
  1. At last, in 1 paragraph, suggest for future works
  • A paragraph concerning possible future work has been added at the end.

Reviewer 4 Report

The manuscript submitted by Sajkiewicz et al. describes the investigation on the surface functionalization of poly(lactide-co-ε-caprolactone) fibres using three different approaches. The area is of importance in the field of scaffolds in tissue engineering applications. The paper is well written, and the approach and the described results are of interest. But, in my opinion, because the issue is the surface functionalization of the fibres and to elucidate the fundamental relationship between functionalisation of the nonwoven material with gelatin and structure of polymer surface, a surface detailed characterization analysis is needed. Because the aim is to modify the outermost layers of the fibers some specific surface characterization analyses is needed. The reported ATR FTIR results refer to the mean composition of the layers of thickness of the order of micron, not exactly a characterization of the outermost layers. For these reasons I would suggest to add surface characterization obtained using techniques such as XPS or SIMS.

Author Response

Review 4

The manuscript submitted by Sajkiewicz et al. describes the investigation on the surface functionalization of poly(lactide-co-ε-caprolactone) fibres using three different approaches. The area is of importance in the field of scaffolds in tissue engineering applications. The paper is well written, and the approach and the described results are of interest. But, in my opinion, because the issue is the surface functionalization of the fibres and to elucidate the fundamental relationship between functionalisation of the nonwoven material with gelatin and structure of polymer surface, a surface detailed characterization analysis is needed. Because the aim is to modify the outermost layers of the fibers some specific surface characterization analyses is needed. The reported ATR FTIR results refer to the mean composition of the layers of thickness of the order of micron, not exactly a characterization of the outermost layers. For these reasons I would suggest to add surface characterization obtained using techniques such as XPS or SIMS.

Answer

Thank you for taking the time to review our work and consider how to improve it.

The research in this work is divided into two main parts. First part consists of examination after surface activation that includes the determination of the presence and amount of functional groups obtained this way on the surface and the assessment of the effect of the activation method on the physical and chemical properties of the material itself. The second part focuses on measuring how much gelatin has been attached and how permanently. Both of these parts require surface testing of the material.

In the first part, two methods of assessing the number of -COOH groups are used - where FTIR-ATR turns out to be a less sensitive method than TBO, the latter, however, gives reliable, consistent results.

In the second part, the presence of attached gelatin is assessed by FTIR-ATR and BCA tests, where FTIR-ATR is used to confirm its presence and BCA gives accurate results on the amount of gelatin.

XPS, as you suggest, could have been used in both parts of this work. We did not run the XPS test for two main reasons. Our previous studies, focusing on aminolysis and gelatin binding, showed that the BCA test was more sensitive and accurate for our samples. The other is that, while we agree with you that FTIR-ATR is not specifically a method used to assessment of surface composition, we believe that it is suitable and adequate to our needs.

It is true that the laser used in FTIR-ATR goes a few microns into the material, but we use it to confirm the presence of gelatin, which in our materials can be found only on the surface, so any result of its detection gives as the answer if it is attached on the surface or not. Later on, we assessed gelatin’s precise amount with BCA, which we believe is a more sensitive method than XPS, and again the fact that gelatin can be found only on the surface, makes BCA a surface assessment method here.

Round 2

Reviewer 1 Report

Accept in present form

Reviewer 4 Report

I would change the title of the article by eliminating the term surface